# Comprehensive Compensation Method for Motion Trajectory Error of End-Effector of Cable-Driven Parallel Mechanism

Yanglong Li [1], Yujun Xue [1,2], Fang Yang [1], Haichao Cai [1] and Hang Li [1,*]

[1]  School of Mechatronics Engineering, Henan University of Science and Technology, Luoyang 471003, China; 200320010015@stu.haust.edu.cn (Y.L.)

[2]  Henan Key Laboratory for Machinery Design and Transmission System, Henan University of Science and Technology, Luoyang 471003, China

[*]  Correspondence: lihang@haust.edu.cn

**Abstract:** The accuracy of the end-effector motion trajectory is a critical performance indicator for cable-driven parallel mechanisms. This study aims to address the problem of trajectory error during the end-effector motion in a cable-driven parallel mechanism. It proposes a comprehensive compensation method based on the simultaneous application of the improved sparrow search algorithm and the cable length space error compensation algorithm, leveraging kinematic analysis. To compensate for the motion trajectory error of the end-effector caused by the geometric parameter error, the study establishes the kinematic model of the cable-driven parallel mechanism using the vector method. It creates the end-effector position error model and motion trajectory error model using the differential kinematic theory, analyzes the impact of the geometric parameter error on the motion trajectory error, constructs the kinematic parameter identification matrix, and uses an improved sparrow search algorithm to compensate for the position error of the motion trajectory interpolation point. For the motion trajectory error of the end-effector caused by non-geometric parameter error, the study analyzes the intrinsic correlation between the adjacent position error of the end-effector and the variation of the cable length using the error similarity theory. It then compensates for the position error of the interpolation point of the trajectory using a cable length space interpolation compensation method to enhance the motion trajectory accuracy of the end-effector. The study experimentally verifies the proposed comprehensive compensation method for end-effector motion trajectory error on a 4-cable-driven 2-DOF parallel mechanism, which reduces the motion trajectory error of the end-effector by 75%.

**Keywords:** cable-driven parallel mechanism; motion trajectory error compensation; parameter identification; improved sparrow search algorithm

## 1. Introduction

Cable-driven parallel mechanism is a special kind of parallel mechanism that has been developed based on rigid parallel mechanisms. They offer several advantages, including a simple mechanical structure, large working space, and strong load capacity [1,2]. As a result, they are used in various fields, such as large equipment handling [3], large radio telescopes [4,5], clinical diagnosis [6], and rehabilitation training [7], among others.

Compared to rigid parallel mechanisms, cable-driven parallel mechanisms use cables instead of rigid linkages to connect the end-effector (also known as the moving platform) to the static platform. Multiple cables are used to control the position, attitude, and trajectory of the end-effector to achieve the desired motion function. Position and attitude control refers to controlling the end-effector at a specified position and attitude angle. The position is usually determined using the coordinate value of the center of mass of the end-effector in a fixed coordinate system. Motion trajectory involves controlling the position of the end-effector to move from the starting point to the endpoint along a specified trajectory.

End-effector motion trajectory control is essential in many applications, such as cable-tracted cameras, cable-driven 3D printing robots, high-speed motion target simulation, and large motion platform simulators [8–11]. These applications have high requirements for motion trajectory accuracy.

The cable-driven parallel mechanism may experience structural dimensional errors during processing and assembly, as well as errors such as elastic deformation of the cable and structural stiffness deformation during operation. These factors can lead to a mismatch between the kinematic or kinetic model parameters established based on the design structure and the actual structure, resulting in model parameter errors. This, in turn, causes position control errors, also known as position errors, where the actual point and the desired position have a coincidence error during the end-effector's motion control. This error causes the actual motion trajectory of the end-effector to deviate from the desired motion trajectory, forming the motion trajectory error. The sources of model parameter errors are mainly two folds: geometric parameter errors caused by the error of the position of the cable exit point and the error of the starting length of the cable, and non-geometric parameter errors caused by the cable's elastic deformation, pulley friction, structural stiffness deformation, and other non-linear factors during the end-effector's movement [12]. These two different types of errors often have a strong non-linear coupling. Thus, to improve the motion trajectory accuracy of the end-effector, it is necessary to eliminate or reduce the influence of both geometric and non-geometric parameter errors on the motion trajectory error.

At present, there is relatively little research literature on end-effector motion trajectory error control. The main method proposed is to correct the model parameter values by kinematic or kinetic model parameter identification of the actual structure to achieve the purpose of eliminating or reducing the model parameter errors.

Zhang et al. [13] proposed an optimal measurement position selection algorithm for kinematic calibration, which used a forbidden search algorithm to search for the measurement position with the optimal observation index. They combined this with a visual measurement system to achieve accurate identification of the geometric parameters of the kinematic model of the cable-driven parallel mechanism. This reduced the mean value of the end-effector trajectory error of the 8-cable 6-DOF cable-driven parallel mechanism from 52.2 mm to 7.1 mm. Wenkai He et al. [14] considered that the cable length error in the cable-driven parallel mechanism is not easy to measure and has the characteristics of force nonlinearity. They proposed a BP neural network-based cable length prediction algorithm by analyzing the kinematic geometric parameter error model of the 6-cable, 6-DOF cable-driven parallel mechanism. This effectively improved the absolute positioning accuracy of the end-effector, but the improvement to the end-effector motion trajectory error was very limited. Li et al. [15] equated two different types of parametric errors, geometric and non-geometric, as pseudo-errors and approximated the end position error profile caused by the pseudo-errors through a neural network. They established a mapping relationship between the end position error and the cable length and compensated for the position error in the joint space. This reduced the mean value of the end-effector trajectory error of the 6-DOF cable-driven parallel mechanism from 7.5 mm to 1.6 mm. Chellal et al. [16] used a combination of identifying kinematic and dynamic parameters of the cable-driven parallel mechanism and adjusting the weighted filter controller to control the end-effector motion. They aimed to improve the motion trajectory tracking accuracy and motion trajectory anti-interference performance of the end-effector and achieve high accuracy trajectory motion of the end-effector of the 6-DOF INCA cable-driven parallel mechanism. Briot et al. [17] identified the kinetic model parameters and joint drive gains of the parallel mechanism based on the least squares method. They experimentally verified the effectiveness of this method on the accuracy of the end-effector trajectory on a 3-DOF orthogonal parallel mechanism. Pengcheng Li et al. [18] proposed a visual closed-loop output error identification method using an optical coordinate measuring machine as the sensor. They considered the highly coupled dynamics of the parallel mechanism and used

non-linear optimization techniques to identify the model parameters. This resulted in a 70% reduction in the root mean square error of the motion trajectory error of the end-effector. In summary, for the motion trajectory error compensation of the end-effector, the existing research mainly analyzes the theoretical research on the trajectory error of the end-effector from the kinematic or kinetic model parameter identification unilaterally, and it is rare to conduct targeted, comprehensive compensation research for the motion trajectory error.

Therefore, this paper presents a comprehensive compensation method for the trajectory error of the end-effector based on the improved sparrow search algorithm and the cable length spatial error compensation algorithm applied simultaneously to improve the trajectory accuracy of the end-effector. This is aimed at the trajectory error problem caused by the geometric and non-geometric parameter errors during the motion of the end-effector of the self-developed 4-cable-driven 2-DOF parallel mechanism for motion target simulation.

## 2. Motion Trajectory Error Modeling

### 2.1. Kinematic Model

The paper describes the working principle of the 4-cable-driven 2-DOF parallel mechanism, illustrated in Figure 1. The mechanism consists of a rectangular vertical frame with four linear motor modules (including linear motors, guides, sensors, etc.) mounted on the frame. The end-effector is fixed in the frame by four cables. The four cables, attached at the center of the end-effector, are connected to four linear motor modules through four pulleys (A1, A2, A3, and A4). By using linear motors to control the change in cable length, the end-effector can change its position within the frame and move along a specified trajectory. Cable length is the distance between the cable exit point and the center of the end-effector. The cable exit point refers to the location of the pulley.

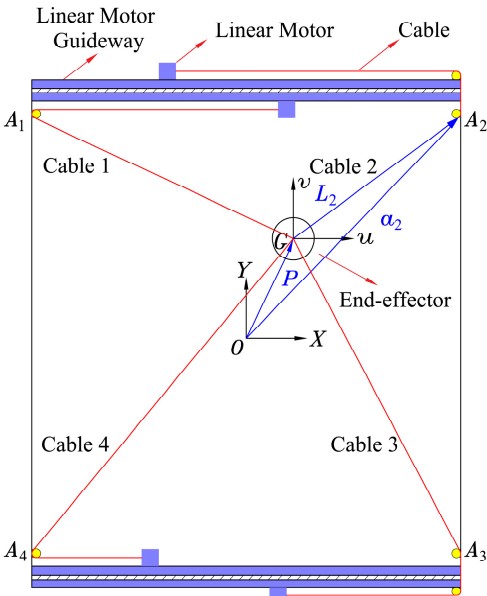

**Figure 1.** The 4-cable-driven 2-DOF parallel mechanism working principle diagram.

Kinematic modeling aims to establish the relationship between the end-effector position space and the cable length space.

As shown in Figure 1, a global coordinate system $O$ is established at the center of the frame plane $O - XY$. The geometric center of the end-effector is denoted by $G$, and the kinematic platform coordinate system $G - uv$ is established, with $G$ as the coordinate origin. The position of the point $G$ in $O - XY$ can be denoted as $P = \begin{bmatrix} x & y \end{bmatrix}^{\mathrm{T}}$, and $A_i$ represents the exit points of the cable $l_i (i = 1, 2, 3, 4)$.

Usually ignoring the elasticity and mass of the cable, assuming that the cable is a rigid cable with no mass. For the $i$th vector closed-loop, the kinematic model of the cable-driven parallel mechanism can be expressed as:

$$L_i = l_i u_i = a_i - P \ (i = 1, 2, 3, 4) \tag{1}$$

where vector $a_i$ represents the position vector of $A_i$ in the world coordinate system (fixed value after the structure size is determined), the position vector of the center of the moving platform $G$ in the world coordinate system is $P$, $L_i = \overrightarrow{GA_i}(i = 1, 2, 3, 4)$ is the vector of cable lengths from point $G$ to $A_i$, $l_i = \|L_i\|$ is the length of the $i$th cable, and $u_i = L_i / l_i$ is the unit vector of each cable.

Considering the relationship between the rate of change of the cable length and the rate of change of the end-effector position, we can obtain the derivative of Equation (1) with respect to time on each side as:

$$\dot{l}_i u_i + l_i \dot{u}_i = \dot{a}_i - \dot{P}(i = 1, 2, 3, 4) \tag{2}$$

Since the value of the coordinates of $a_i$ in a fixed coordinate system does not change with time, we can simplify Equation (2) by left multiplying both sides by $u_i^{\mathrm{T}}$ to obtain:

$$\dot{l}_i = -u_i^{\mathrm{T}} \begin{bmatrix} \dot{x} \\ \dot{y} \end{bmatrix} \tag{3}$$

The linear velocity vector $v = \begin{bmatrix} \dot{x} & \dot{y} \end{bmatrix}^{\mathrm{T}}$ of the dynamic platform coordinate system with respect to the world coordinate system can be simplified using Equation (3) to obtain Equation (4).

$$\dot{l} = -Jv \tag{4}$$

where $\dot{l} = \begin{bmatrix} \dot{l}_1, \dot{l}_2, \dot{l}_3, \dot{l}_4 \end{bmatrix}^{\mathrm{T}}$ is the rate of change of the cable length and $J$ represents the velocity Jacobi matrix of the cable-driven parallel mechanism.

$$J = \begin{bmatrix} u_1^{\mathrm{T}} \\ u_2^{\mathrm{T}} \\ u_3^{\mathrm{T}} \\ u_4^{\mathrm{T}} \end{bmatrix} \tag{5}$$

A 4-cable-driven 2-DOF parallel mechanism has a specific set of structural parameters that can be used to derive a 4-row, 2-column Jacobi matrix $J$. This matrix establishes the relationship between the end-effector workspace and the cable length space of the cable-driven parallel mechanism. It maps the differential of each cable length in the same time period to the differential of the end-effector position coordinates.

### 2.2. Position Error Model

Position error modeling aims to establish the relationship between errors in geometric parameters, such as cable exit point position error and cable length error, and the resulting errors in end-effector position.

The position error of the end-effector can be expressed as:

$$dP = \begin{bmatrix} dp_x, dp_y \end{bmatrix}^{\mathrm{T}} \tag{6}$$

where $dp_x$, $dp_y$ are the position error coordinates of the end-effector in the world coordinate system.

By taking the same differentiation for both sides of Equation (1) and multiplying it by $u_i^{\mathrm{T}}$, Equation (7) is obtained:

$$u_i^{\mathrm{T}} dl_i u_i + u_i^{\mathrm{T}} l_i u_i = u_i^{\mathrm{T}} da_i - u_i^{\mathrm{T}} dP \tag{7}$$

Using the theory of differential kinematics [11], Equation (7) can be simplified as:

$$dl_i = u_i^{\mathrm{T}} da_i - u_i^{\mathrm{T}} dP \tag{8}$$

Simplifying Equation (8) further yields:

$$dl = -JdP + J_e da \tag{9}$$

where $J$ is the Jacobi matrix and $J_e$ is the vector along the cable direction.

Left-multiplying both sides of the Equation (9) by $J^+$ ($J^+ = (J^{\mathrm{T}} J)^{-1} J^{\mathrm{T}}$ is the generalized inverse of $J$) and substituting $dP$, $da$, $dl$ for $\delta e$, $\delta a$, $\delta l$, respectively, yields the end-effector position error model as:

$$\delta e = J^+ (J_e \delta a - \delta l) \tag{10}$$

For the 4-cable-driven 2-DOF parallel mechanism studied in this paper, Equation (10) can be expanded as follows:

$$\begin{bmatrix} \delta P_x \\ \delta P_y \end{bmatrix} = \left( \begin{bmatrix} u_1^{\mathrm{T}} \\ u_2^{\mathrm{T}} \\ u_3^{\mathrm{T}} \\ u_4^{\mathrm{T}} \end{bmatrix}^{\mathrm{T}} \begin{bmatrix} u_1^{\mathrm{T}} \\ u_2^{\mathrm{T}} \\ u_3^{\mathrm{T}} \\ u_4^{\mathrm{T}} \end{bmatrix} \right)^{-1} \begin{bmatrix} u_1^{\mathrm{T}} \\ u_2^{\mathrm{T}} \\ u_3^{\mathrm{T}} \\ u_4^{\mathrm{T}} \end{bmatrix}^{\mathrm{T}} \left( \begin{bmatrix} u_1^{\mathrm{T}} & & & \\ & u_2^{\mathrm{T}} & & \\ & & u_3^{\mathrm{T}} & \\ & & & u_4^{\mathrm{T}} \end{bmatrix} \begin{bmatrix} \delta a_1 \\ \delta a_2 \\ \delta a_3 \\ \delta a_4 \end{bmatrix} - \begin{bmatrix} \delta l_1 \\ \delta l_2 \\ \delta l_3 \\ \delta l_4 \end{bmatrix} \right) \tag{11}$$

*2.3. Motion Trajectory Error Model*

The cable-driven parallel mechanism's structure indicates that the end-effector trajectory is a function of the kinematic model parameters. As the cable length $l(l = l(t))$ changes over time, the end-effector position $P$ moves along the desired trajectory to form the end trajectory $F$. The desired trajectory of the end-effector can be expressed as:

$$F = f(a, l) \tag{12}$$

where $a$ and $l$ are the nominal vectors of kinematic parameters for the cable-driven parallel mechanism.

Based on the previous analysis, it is evident that the actual trajectory of the end-effector will deviate from the desired trajectory due to geometric parameter errors such as cable exit point position error and cable length error, as shown in Figure 2. The actual trajectory $F_r$ of the end-effector can be expressed as:

$$F_r = f(a + da, l + dl) \tag{13}$$

The motion trajectory error can be expressed as:

$$\Delta F = F_r - F \tag{14}$$

At moment $t_j (j = 1, 2, \ldots, n)$, the distance between the actual position $P'_j$ and the theoretical position $P_j$ on the motion trajectory of the end-effector is given by Equation (15):

$$d_0^2 = (F_r - F)^{\mathrm{T}} (F_r - F) = \Delta F^{\mathrm{T}} \Delta F \tag{15}$$

Combining Equations (11) and (15) gives Equation (16), which shows how to calculate the processing of the position error model.

$$d_0^2 = \delta e^{\mathrm{T}} \delta e \tag{16}$$

where $\delta e$ is the error value of the corresponding position on the trajectory of the end-effector at moment $t_j$.

Equation (17) is obtained when the motion trajectory error $\|\Delta F\|$ is expressed as the average of the position errors of $m$ end-effector position measurement points in the motion trajectory.

$$\|\Delta F\| = \sum_{i=1}^{m} \delta e^{\mathrm{T}} \delta e / m = \sum_{i=1}^{m} \sqrt{\delta p_x^2 + \delta p_y^2} / m \tag{17}$$

The analysis of Equation (17) shows that in order to make the actual trajectory of the end-effector close to the desired trajectory, the value of the trajectory error $\|\Delta F\|$ of the end-effector should tend to the minimum value, that is, Minimize $\|\Delta F\|$.

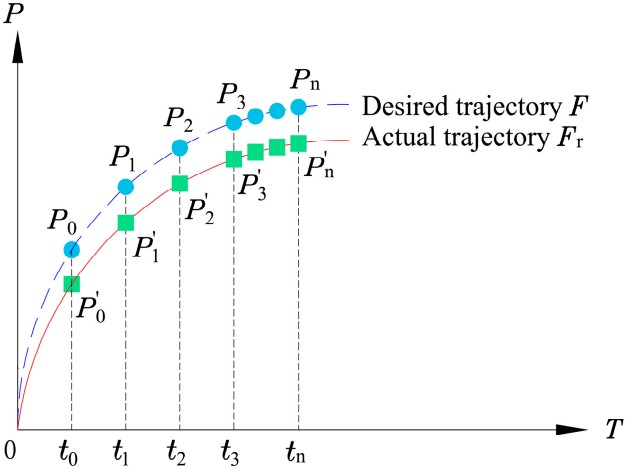

**Figure 2.** Motion trajectory under the influence of geometric parameter errors.

## 3. Motion Track Accuracy Compensation

### 3.1. Geometric Parameter Error Identification and Analysis

Parameter identification is required first to eliminate the motion trajectory errors of the end-effector using the geometric parameter error compensation method. This is achieved by constructing a kinematic parameter identification matrix to identify the geometric parameter error. Once identified, the geometric parameter error compensation method can be used to eliminate the motion trajectory error of the end-effector.

In the workspace of the cable-driven parallel mechanism, the matrix $J$ is non-singular, and the identification model can be obtained by rewriting Equation (10) as:

$$\delta e = J_m \delta p \tag{18}$$

where $\delta p = \begin{pmatrix} \delta a & \delta l \end{pmatrix}^{\mathrm{T}}$ is the geometric parameter error, $J_m$ is the error Jacobi matrix

$$J_m = \begin{bmatrix} J^+ J_e & -J^+ \end{bmatrix} \tag{19}$$

The geometric parameter error $\delta p$ is usually obtained by solving the identification model using the least squares method. The least squares solution can be calculated as follows:

$$\delta p = \left( J_m^{\mathrm{T}} J_m \right)^{-1} J_m^{\mathrm{T}} \delta e \tag{20}$$

The coordinate values of the measurement points on the $m$ end-effector can be obtained through actual measurement in the field. These points can also be referred to as trajectory interpolation points.

Based on the analysis of Equation (10), the end-effector position error model can be derived. It shows that the cable exit point error $\delta a$ cannot be directly compensated while $\delta a$ can be converted into $J_e \delta a$, which is a more easily measurable and controllable cable length error, that is, $\delta l' = J_e \delta a$. Consequently, Equation (10) can be simplified as:

$$\delta e = J^+ (J_e \delta a - \delta l) = J^+ (\delta l' - \delta l) \tag{21}$$

If we let $(\delta l' - \delta l) = \delta l''$, then Equation (21) can be further simplified as:

$$\delta e = J^+ \delta l'' \tag{22}$$

Equation (22) shows that adjusting the cable length parameter $\delta l''$ can compensate for the position error of the end-effector $\delta e$. Therefore, optimizing the cable length parameter $\delta l''$ to compensate for the error at each position on the end-effector's motion trajectory can achieve error compensation for the end-effector motion trajectory.

The motion trajectory error model of the end-effector is a multi-input and multi-output model. In this paper, we use the end-effector trajectory error as the objective function and aim to compensate for the end-effector trajectory error caused by the geometric parameter error by improving the sparrow search algorithm to find the optimal cable length parameter $\delta l''$.

*3.2. Compensation of Motion Trajectory Error Based on Improved Sparrow Search Algorithm*

3.2.1. Improving Sparrow Search Algorithm

The Sparrow Search Algorithm (SSA) is an intelligent optimization algorithm that draws inspiration from the foraging and anti-predator behaviors of sparrow populations [19]. SSA has several advantages, such as good stability, strong global search ability, and few parameters. However, it is susceptible to falling into local optima during the late stages of population evolution.

To improve the application effect of this algorithm, this paper introduces dynamic weights $\omega$ to balance global search and local search and prevent the algorithm from converging into local optima. As a result, the authors propose an Improved Sparrow Search Algorithm (ISSA) that utilizes dynamic adaptive weights.

The expression of the dynamic weight $\omega$ is as follows:

$$\omega = \omega_{\max} - \frac{(\omega_{\max} - \omega_{\min})t}{T} + (0.5 - q)\left(1 - \frac{t}{T}\right)^2 \tag{23}$$

where $\omega_{\max}$ and $\omega_{\min}$ are the preset maximum and minimum weight coefficients set beforehand, $T$ is the maximum number of iterations, $t$ is the current number of iterations, and $q$ follows a uniform distribution within the interval $[0, 1]$.

The sparrow search algorithm's optimized search mechanism finder position update formula, combined with the above dynamic weights $\omega$, defined above, is as follows:

$$x_{i,d}^{t+1} = \begin{cases} x_{i,d}^t + \omega \cdot \left(x_{bestd}^t - x_{i,d}^t\right), R_2 < ST \\ x_{i,d}^t + Q \cdot L, R_2 \geq ST \end{cases} \tag{24}$$

where $t$ is the number of iterations, $Q$ is a random number with standard normal distribution, $L$ is a matrix with $1 \times d$ elements of 1, $R_2 \in [0, 1]$ and $ST \in [0.5, 1]$ represent the warning value and safety threshold, respectively. $x_{id}^t$ is the position of the $i$th sparrow in the $d$th dimension at the number of iterations $t$, and $x_{bestd}^t$ is the optimal position of the sparrow in the $d$th dimension at the $t$th generation.

Based on the motion trajectory error model in Section 2.3, we established the motion trajectory error compensation objective function, that is, the adaptation degree function in ISSA:

$$\Phi_{\min} = \sqrt{\delta p_x^2 + \delta p_y^2} \tag{25}$$

The process of compensating the motion trajectory error of the end-effector using the ISSA seeking solenoid length parameter is shown in Figure 3.

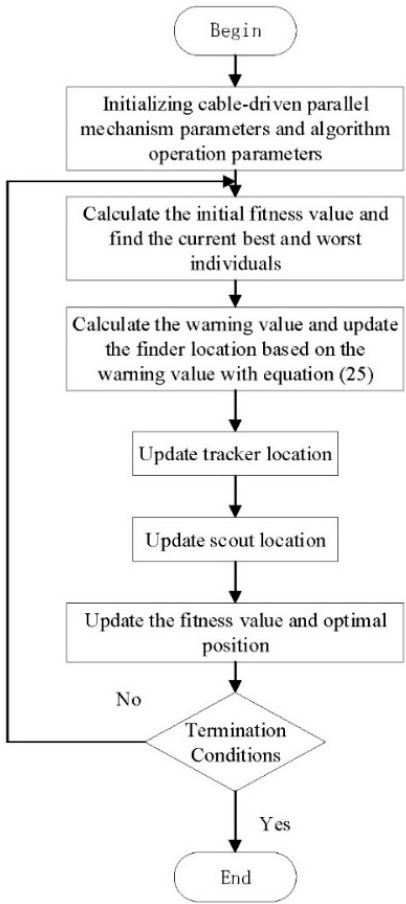

**Figure 3.** ISSA trajectory error compensation flow.

3.2.2. Motion Trajectory Error Compensation Simulation Based on ISSA Algorithm

The ISSA and SSA are simultaneously applied to the end-effector motion trajectory error compensation to verify the effectiveness of the ISSA-based motion trajectory error compensation algorithm and to conduct comparison tests. The end-effector trajectory task is set in the workspace of the cable-driven parallel mechanism: with the parameters, $x = 200 \sin(0.8\pi t)$, $y = 200 \cos(0.8\pi t)$, time of 2.5 s, and step length of 0.05 s. The initial parameters of the ISSA and SSA are set with the sparrow population size $Sizepop = 30$, a maximum number of evolution $maxgen = 500$, and an initial number of discoverers $rNum = 0.2 \times Sizepop$, and the number of scouts at 20% of the population size. The weight coefficients used in the fitness function are $\omega_{\max} = 0.9$, $\omega_{\min} = 0.4$, and the search space of the sparrow is $l_{iMin} \leq l_i \leq l_{iMax}$ ($i = 1, 2, 3, 4$). The simulation results show that the trajectory error of the end effector changes with the number of ISSA and SSA iterations, as shown in Figure 4.

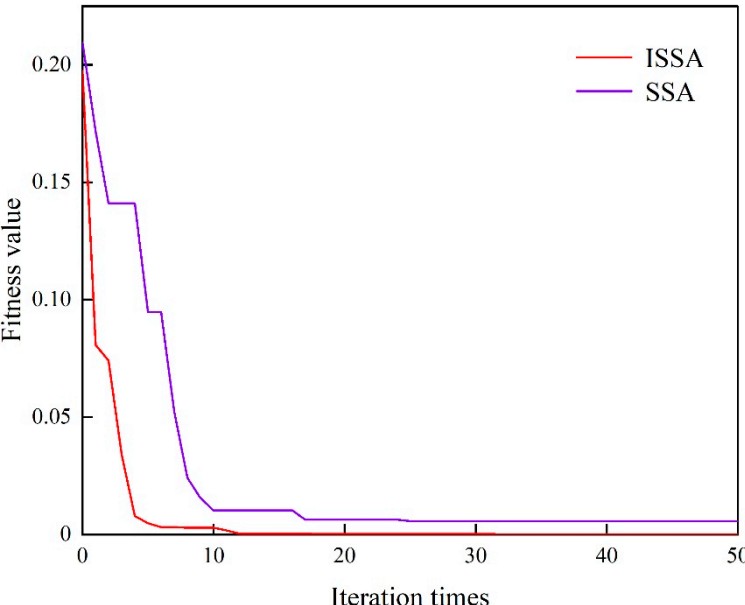

**Figure 4.** Variation curve of motion trajectory error with the number of ISSA/SSA iterations.

As shown in Figure 4, The compensation effect of the ISSA is significantly better than that of the SSA, which falls into the local optimum during the iterative process of the SSA. Meanwhile, the ISSA cannot only quickly jump out of the local optimum and continue iteration, but also the end-effector trajectory error gradually decreases and converges to zero with the increase of iterations. Specifically, the end-effector trajectory error decreases rapidly in the first five iterations of ISSA, and after the 13th iteration, it stabilizes and gradually converges to the global optimum. These simulation results demonstrate that the ISSA proposed in this paper can converge to the global optimum and effectively improve the accuracy of the end-effector's motion trajectory.

Figure 5 depicts the trajectory of the end-effector's motion before and after ISSA compensation. Meanwhile, Figures 6 and 7 illustrate the distribution of the end-effector's trajectory error along the coordinate axis direction before and after ISSA compensation. In Figure 6, the maximum value of the end-effector trajectory error is 1.365 mm in the $x$-direction and 1.376 mm in the $y$-direction. Moreover, the end-effector trajectory error exhibits a triangular function-like distribution along the $x$-direction and the $y$-direction.

On the other hand, Figure 7 shows the error distribution of the motion trajectory of the end-effector along the coordinate axis after ISSA compensation. The maximum error value in the $x$-direction is $4.683 \times 10^{-14}$ mm, and the maximum error value in the $y$-direction is $8.605 \times 10^{-14}$ mm. The effect of ISSA compensation is evident, and the influence of the geometric parameter error on the motion trajectory of the end-effector is practically eliminated.

Before compensating for the motion trajectory error, the geometric parameter error was the factor affecting the motion trajectory. This error was a fixed value, which was included in the motion trajectory error model discussed in Section 2.3. It can be observed that the motion trajectory error $\|\Delta F\|$ varies with the error Jacobi matrix $J_m$. Moreover, the value of the Jacobi matrix is associated with the end-effector position. Therefore, if the motion trajectory task along the coordinate axis direction expression is a triangular function, the motion trajectory error along the coordinate axis direction will also have a trigonometric-like distribution.

With ISSA compensating for the motion trajectory error, the ISSA seeks the optimal fitness value, which varies with the global optimal value search. However, the error Jacobi matrix $J_m$ is constant. Therefore, the end-effector motion trajectory error distribution fluctuates irregularly after ISSA compensation, as depicted in Figure 7.

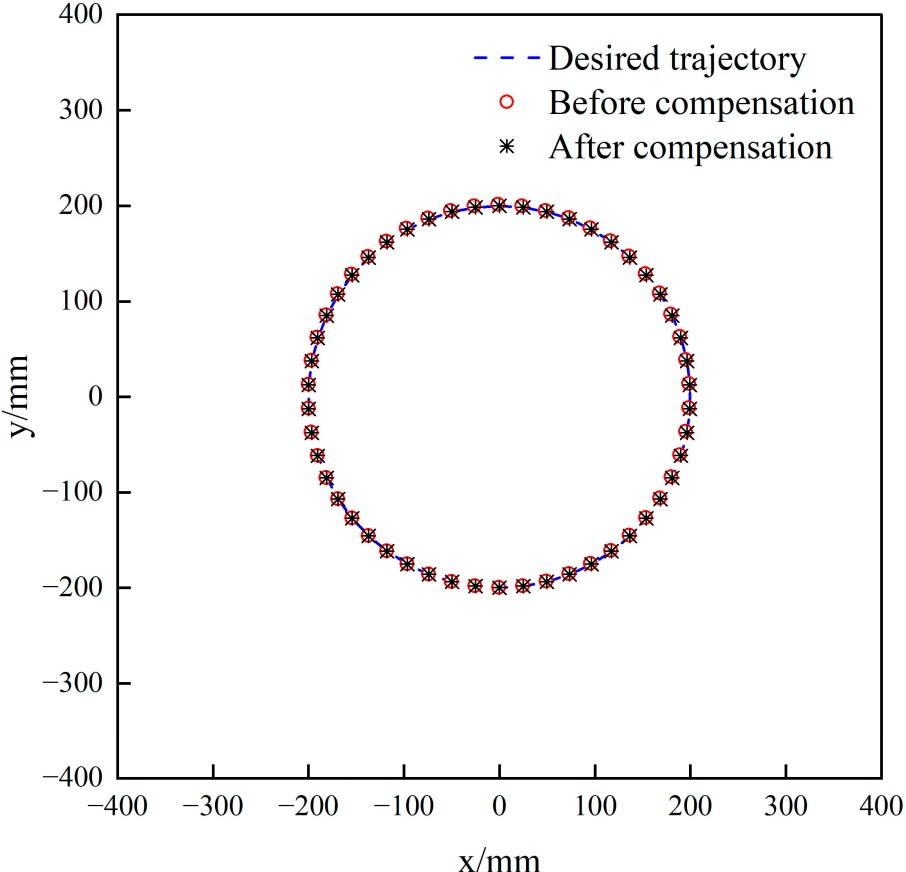

**Figure 5.** End-effector motion trajectory.

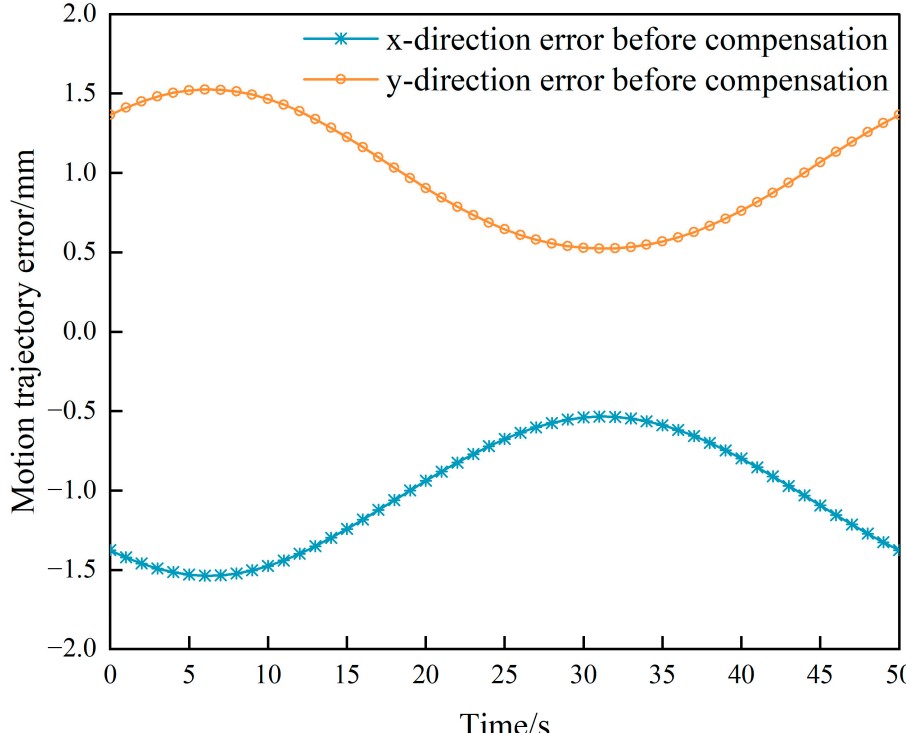

**Figure 6.** End-effector motion trajectory error before ISSA compensation.

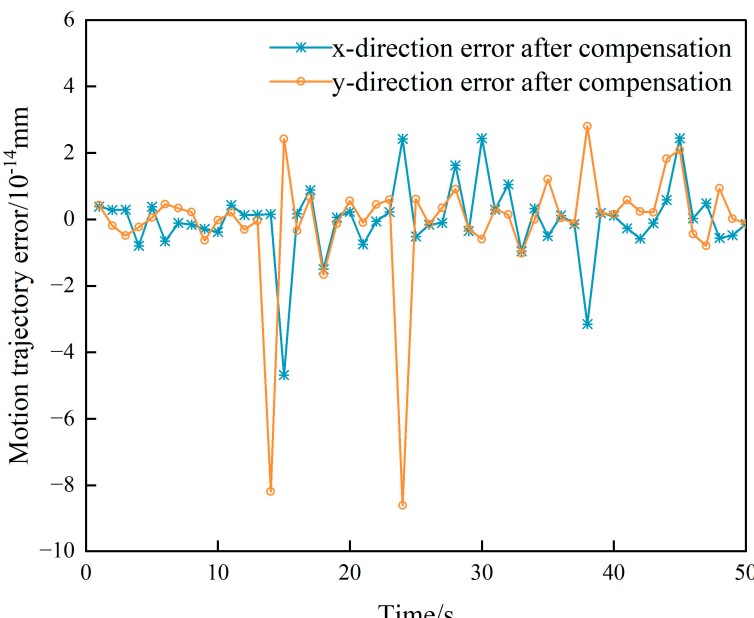

**Figure 7.** End-effector trajectory error after ISSA compensation.

### 3.3. Interpolation Error Compensation in the Space of the Cable Length

During the operation of a cable-driven parallel mechanism, the elasticity of the cable causes elongation, leading to a cable length error. This error is the main influencing factor of the non-geometric parameter errors. In this paper, we propose the use of a spatial interpolation error compensation algorithm to counteract the motion trajectory error caused by the elastic elongation of the cable.

By combining the position error similarity theory [20] with the kinematic model, position error model, and trajectory error model established in Equations (1), (10) and (17), we can analyze the system. In the kinematic model Equation (1), the position $P$ of the end-effector is determined by a set of corresponding cable length parameters $l = [l_1, l_2, l_3, l_4]$; the cable and changes in cable length correspond to changes in the end-effector position. In the position error model m Equation (9), the changes in cable length parameters and end-effector positions of the cable-driven parallel mechanism are continuously differentiable after fully differentiating the kinematic model. This means that when there are micro-elementary changes in the errors of the cable length parameters, there are corresponding micro-elementarily changes in the end-effector positions. Therefore, the corresponding error curves for the two are smooth and continuous.

Based on the theory of position error similarity, we know that when the Euclidean distance between the two sets of cable length parameters of the cable-driven parallel mechanism is close to each other, their position error vectors have a certain degree of similarity. In other words, this means that when the Euclidean distance between two groups of cable lengths reaches a certain degree of similarity, one group of cable length errors can be used to replace the other group of cable length errors.

Based on the unique solutions of the forward and inverse kinematics of the 4-cable-driven 2-DOF parallel mechanism, as well as the continuity of the cable length error and the force state and deformation of the mechanism, we propose an interpolation error compensation in the space of the cable length algorithm for cable-driven parallel mechanism. This algorism compensates for the motion trajectory error of the end-effector by using the space division grid interpolation method [21], and the steps are as follows:

1. A number of specified positions are obtained between the start and end points of the end-effector trajectory based on a given step size. These positions are also known as trajectory interpolation points, and their coordinate values are denoted by $(x, y)$. The kinematic inverse solution is used to obtain the cable length value $l = [l_1, l_2, l_3, l_4]$ corresponding to the trajectory interpolation points;

2. The workspace of the cable-driven parallel mechanism is divided into a series of positive quadrilateral grids based on the space division grid interpolation method. Suppose the coordinates of the theoretical position of the grid vertex of the square quadrilateral in which the trajectory interpolation point is located are $(x_{tK}, y_{tK})$. Then, the corresponding theoretical cable length of the grid vertex in the cable length space is $l_{tK} = [l_{tK,1}, l_{tK,2}, l_{tK,3}, l_{tK,4}]$ $(K = 1, 2, \ldots, 4)$. The four vertices of the quadrilateral where the trajectory interpolation points are located are used as "sampling points." The actual position coordinates $(x_{trK}, y_{trK})$ of the sampling points of the mechanism are measured by measuring instruments, and the kinematic inverse solution model is used to solve the error of the cable length corresponding to the movement of the end-effector to each point:

$$\Delta l_{tK} = [\Delta l_{tK,1}, \Delta l_{tK,2}, \Delta l_{tK,3}, \Delta l_{tK,4}] \tag{26}$$

where, $\Delta l_{tK,i} = \Delta l_{trK,i} - \Delta l_{tK,i}$, $i = 1, 2, \ldots, 4$, $K = 1, 2, \ldots, 4$.

3. The error compensation for the end-effector trajectory interpolation points is evaluated using the cable length spatial distance evaluation function. However, since the trajectory interpolation point is typically not located precisely on a grid vertex during the motion of the end-effector, an interpolation error compensation method based on the inverse distance weight function of the cable length space is proposed in this paper. This method takes into account the continuity of the error of the cable-driven parallel mechanism; it does not happen to be on the grid vertex. Therefore, combined with the above-mentioned characteristics of continuity of the error of the cable-driven parallel mechanism, this paper proposes an interpolation error compensation method based on the inverse distance weight function of the cable length space and enables the calculation of the cable length space error value of the trajectory interpolation point. This value is then used to adjust the driving cable length of the trajectory interpolation point, compensating for the motion trajectory error of the end-effector caused by non-geometric parameters.

The distance weights $w_K$ of the $K$th vertex of the quadrilateral with respect to the interpolation point of the trajectory can be calculated using Equation (27):

$$w_K = \frac{\left(\frac{1}{d_K}\right)^r}{\sum\limits_{K=1}^{4} \left(\frac{1}{d_K}\right)^r} \tag{27}$$

Here, $d_K = \|l_{tK} - l\|_2^2$, represents the cable length distance in space and $r$ is the weighted power exponent, which is set to 1 in this case. The value of $\sum\limits_{K=1}^{4} w_K = 1$ is derived from Equation (27).

Using the weight $w_K$ and the cable length space error of the trajectory interpolation point and the end-effector at each vertex, the cable length error value of the trajectory interpolation point can be calculated as shown in Equation (28):

$$\Delta l_i = \sum_{K=1}^{4} w_k \Delta l_{tK,i} (i = 1, 2, 3, 4) \tag{28}$$

The cable length error value of the trajectory interpolation point is used to compensate for the error in the cable length of the trajectory interpolation point to reach the desired cable

length $l_d$. This compensation ensures that the actual drive cable length of the compensated trajectory interpolation point of the end-effector trajectory is obtained as follows:

$$l_c = l_d + \Delta l \tag{29}$$

where $\Delta l = [\Delta l_1, \Delta l_2, \Delta l_3, \Delta l_4]$.

## 4. Experimental Verification

### 4.1. Experimental Device

The self-developed 4-cable-driven 2-DOF parallel mechanism was experimentally verified for its end-effector motion trajectory accuracy compensation.

Figure 8 shows the physical diagram of the 4-cable-driven 2-DOF parallel mechanism used in the experiment. The static platform is made of aluminum alloy and has a rectangular frame measuring 1.5 m × 2.1 m, which is assembled by aluminum profiles to provide a benchmark for the installation and positioning of other components. The guide pulley group consists of a guide pulley, preload pulley, and pulley bracket to provide guidance and preload for the cable. The end-effector, which is the dynamic platform, is an aluminum hollow cylinder of uniform mass weighing mass is about 0.3 kg, which is the main driving target of the cable-driving parallel mechanism. Four linear motors consisting of linear motors, guideways, sensors, etc., provide the driving force for the cable-driven parallel mechanism. The drive modules are mounted on the top and bottom sides of the frame. Four cables drawn by the drive modules are crossed and fastened to the geometric center of the end-effector diagonally. A special red visual target is fixed on the end-effector, and a Balser high-speed camera-based vision measurement system is used to track and capture the circle center of the target, which is used as a point feature to obtain the position and motion trajectory information of the end-effector.

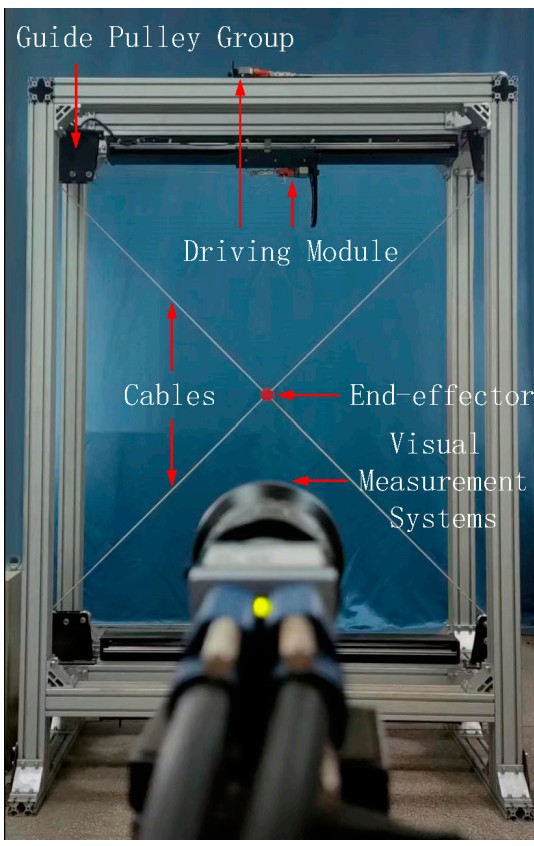

**Figure 8.** Physical diagram of the 4-cable-driven 2-DOF parallel mechanism.

The vision measurement system consists of a camera, an image acquisition card, and an industrial control PC. The camera completes the image acquisition work, and the industrial control PC completes the image processing, feature extraction, and position calculation. The maximum sampling frequency of the adopted vision measurement system is 340 frams/s, the average value of position measurement error is 0.199 mm, and the standard deviation is 0.0263 mm, which can meet the position tracking measurement requirement of the end-effector.

### 4.2. Experimental Method

The end-effector motion trajectory accuracy compensation experiments were conducted following the guidelines outlined in the international standard ISO 9283:1998 [22] for industrial robot trajectory accuracy testing. According to this standard, industrial robot trajectory accuracy is determined by the deviation between the position of the commanded trajectory and the position of each actual trajectory to the centerline of the trajectory position cluster. The trajectory accuracy is expressed as $AT_p$:

$$AT_p = \sqrt{(\overline{x}_i - x_i)^2 + (\overline{y}_i - y_i)^2 + (\overline{z}_i - z_i)^2} \tag{30}$$

Here $\overline{x}_i$, $\overline{y}_i$ and $\overline{z}_i$ are the average values of the interpolated points' coordinates of the given motion trajectory in all directions. $i = 1, \ldots, m$, are the number of interpolated points of the given motion trajectory.

The specific experimental procedures were as follows:

1.  Based on the selection of the experimental space in the $AT_p$ experiment and the characteristics that the working space of the studied 4-cable-driven 2-DOF parallel mechanism is planar, a square quadrilateral with a side length of 520 mm in the working space is selected to test the accuracy of the end-effector trajectory, and the world coordinates system $O - XY$ of the cable-driven parallel mechanism is set at the center of the quadrilateral, and the coordinate of the vertex $C_1$ in the world coordinate system $O - XY$ is $(-260, 260)$.

2.  The line segment $P_2P_3 = 0.8 \times C_1C_3$ was selected as the experimental trajectory for the end-effector motion trajectory accuracy test. Fifty trajectory interpolation points were chosen on the trajectory, and the grid where the trajectory interpolation points are located was divided according to a certain grid size, as shown in Figure 9. In the figure, $A_1 \sim A_5$ are the five trajectory interpolation points, and the pink quadrilateral is the grid corresponding to the trajectory interpolation points, and the position errors of the grid vertices need to be measured to compensate for the non-geometric parameter errors in the motion trajectory errors when the interpolation error compensation algorithm is applied in the cable length space;

3.  A vision measurement system was used to measure the trajectory error and compensate for the end-effector at 50 trajectory interpolation points. The numerical changes of $AT_p$ before and after the algorithm compensation were compared to evaluate the compensation effect of the algorithm.

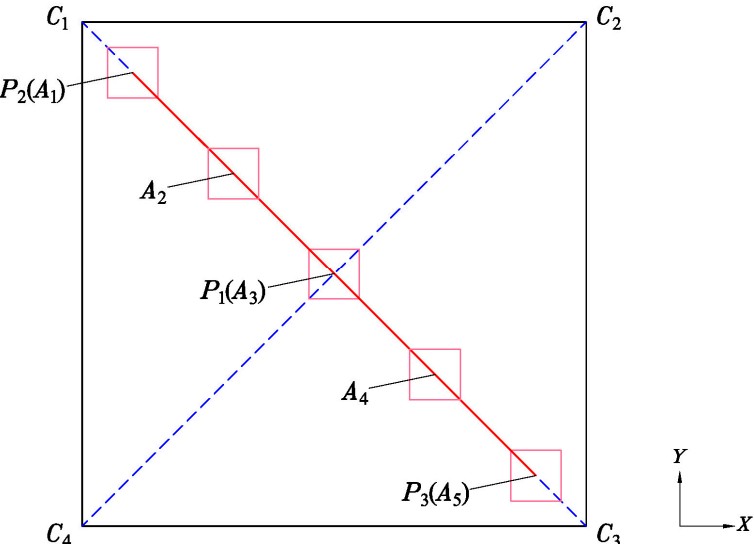

**Figure 9.** Experimental trajectory.

*4.3. Compensation of Motion Trajectory Error Based on Improved Sparrow Search Algorithm Experimental Results*

Fifty static position points in the workspace of the cable-driven parallel mechanism were selected, and the position errors were measured using the visual measurement system. The experimental results of the position error for 50 static positions of the end-effector before and after ISSA compensation are shown in Figures 10 and 11. The maximum value, mean value, and variance of the position error of the end-effector before the ISSA compensation are 10.064 mm, 4.228 mm, and 4.461 mm, respectively. The maximum value, mean value, and variance of the position error of the end-effector after ISSA compensation are 4.275 mm, 2.568 mm, and 0.758 mm, respectively. The position error of the end-effector is reduced by 42.4% after ISSA compensation, which effectively compensates for the position error of the end-effector.

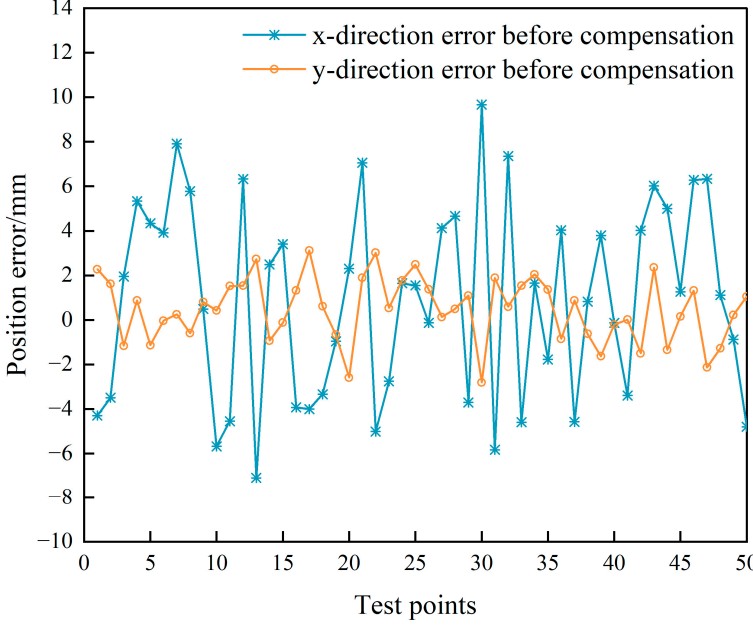

**Figure 10.** Position error before compensation.

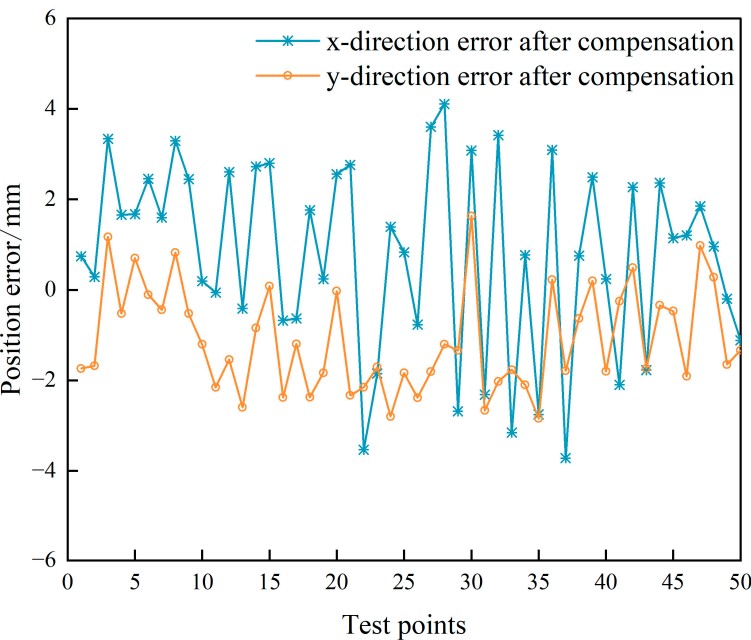

**Figure 11.** Position error after compensation by the optimization algorithm.

The trajectory accuracy ($AT_p$) was used to evaluate the motion trajectory error of the end-effector given motion trajectory $P_2P_3$ before and after the ISSA compensation. The results showed that before compensation, the trajectory accuracy was 5.949 mm, while after compensation, it improved to 2.625 mm. The motion trajectory error of the end-effector was reduced by 55.9% compared to that before ISSA compensation. The following will further compensate for the motion trajectory error of $P_2P_3$ using the method of interpolation error compensation in the space of cable length.

*4.4. Experimental Results of Interpolation Error Compensation in the Space of Cable Length*

The experimental plane was divided into 40 mm, 20 mm, and 10 mm grids. The visual measurement system was used to measure the error value of each grid vertex and calculate the compensation value of the cable length for 50 interpolated points of the given motion trajectory $P_2P_3$ of the end-effector by combining with Equation (28), thus compensating the accuracy of the motion trajectory in the cable length space.

The numerical value of the trajectory accuracy ($AT_p$) was used to evaluate the error compensation effect of different grid sizes on the motion trajectory $P_2P_3$. The results showed that when the grid sizes were 40 mm, 20 mm, and 10 mm, the corresponding $AT_p$ reduced from 2.625 mm to 1.791 mm, 1.569 mm, and 1.548 mm, respectively. The analysis of the results indicated that as the mesh division size decreased, the motion trajectory error after the interpolation compensation of the cable length space decreased, and this also increased the measurement workload of the mesh vertex position information. Moreover, the marginal effect of the motion trajectory error compensation effect decreased and became less significant.

**5. Conclusions**

In this study, we aim to address the trajectory errors that arise from geometric and non-geometric parameter errors during the motion of the end-effector of a 4-cable-driven 2-DOF parallel mechanism. The mechanism was developed independently for motion target simulation. To mitigate the trajectory error of the end-effector motion, we proposed a comprehensive compensation method. Our approach is based on the simultaneous application of the improved sparrow search algorithm and the cable length space error compensation algorithm.

A model of kinematics of the end-effector parallel mechanism is established by the vector method, and a model of position error and a model of motion trajectory error of the end-effector is established by differential kinematics theory. Based on the position error model, the influence of geometric parameter error on the motion trajectory error is analyzed, and it is found that the optimized cable length parameter can reduce the influence of geometric parameter on the trajectory accuracy. An improved sparrow search algorithm is used to compensate for the position error of the motion trajectory interpolation point, and the motion trajectory error compensation is realized. The correctness of the algorithm was verified using experiments and simulations, respectively.

To address the end-effector trajectory errors caused by non-geometric parameter errors, the intrinsic correlation between the adjacent position errors of the end-effector and the variation of the cable length is analyzed by the error similarity theory. It is found that the end-effector position errors have the characteristics of continuity in the cable length space, the trajectory interpolation point position errors are compensated by a cable length space interpolation compensation method, and the actual trajectory drive value of each trajectory interpolation point is solved by using the cable length space distance weight function, which improves the motion trajectory accuracy of the end-effector.

Experimental validation results demonstrate that the improved sparrow search algorithm to compensate for geometric parameter error reduces motion trajectory error from 5.949 mm to 2.625 mm. Similarly, using the interpolation error compensation in the space of the cable length algorithm further reduces motion trajectory error from 2.625 mm to 1.548 mm. By integrating both algorithms simultaneously, motion trajectory error is reduced by 75%.

**Author Contributions:** Conceptualization, Y.L. and H.L.; methodology, Y.L.; software, Y.L.; validation, Y.L., H.L., Y.X., F.Y. and H.C.; formal analysis, Y.L.; investigation, Y.L.; resources, Y.L.; data curation, F.Y.; writing—original draft preparation, Y.L.; writing—review and editing, H.L. and Y.L.; visualization, Y.L. and H.L.; supervision, H.L. and Y.X.; project administration, H.C.; funding acquisition, Y.X. All authors have read and agreed to the published version of the manuscript.

**Funding:** This research was funded in part by the National Basic Scientific Research Project of China, grant number JCKY2019419D001; in part by the Scientific and Technological Key Project of Henan Province, grant number 202102210078; and in part by the Key Scientific Research Projects of Colleges and Universities of Henan Province, grant number 19A460017.

**Data Availability Statement:** The data generated during the current study are available from the corresponding author on reasonable request.

**Conflicts of Interest:** The authors declare no conflict of interest.

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
