# Peer review of "Comprehensive Compensation Method for Motion Trajectory Error of End-Effector of Cable-Driven Parallel Mechanism"

_machines, doi:10.3390/machines11050520_

Round 1
Reviewer 1 Report
This paper presents a compensation method based on the sparrow search algorithm to improve the motion trajectory in the end-effector in the cable-driven parallel mechanism.
In general, it includes a good description of the proposed methods, and the document is easy to read and understand. However, I have the following suggestions or questions to abord in the document:
Page 4: Although the steps to go from equation 3 to equation 4 are described, it is not clear to me how equation 4 is obtained.
In the improved sparrow search algorithm, the parameter (0.5-q) is not clear. Some lines after this equation, you mention the warning value. Is it? How affect the variation of this parameter to the results? According to the results, improvements of 50% are obtained, is this directly related to this parameter? I suggest to include a figure or a table to show how to affect this parameter to the final results.
Figure 3 has mixed languages (English and Chinese).
Although is not the aim of this paper to reduce the rapid transitions in the error changes observed in Figures 7 and 11, is it possible to smooth these transitions, or can you explain if these transitions can generate not desired movements in real applications?
Reviewer 2 Report
The paper does not clearly presents the contribution or novelty of the paper. The introduction includes previous works, however, the novelty is not exactly expressed. By this way the paper is not acceptable.
The conclusion is a little bit general. The contributions of the paper should be more clearly presented.
Author Response
Dear review expert, Please see the attachment.

Reviewer 3 Report
The work is interesting to be read and the research presents some level of novelty.
It proposes a comprehensive compensation method based on improved sparrow search algorithm and the cable length space error compensation algorithm to compensate for the motion trajectory error of the end-effector caused by the geometric parameter error.
However, there are some observations that should be made:
- it is not correct to say that the robots on cables are newly developed, they have been present for decades;
- the cables must be considered inextensible, so the notion of flexible cable not be used. Indeed, their elastic deformation must be taken into consideration.
- position also means altitude and trajectory, one needs explanations related the terms "position and altitude angle". The position is determined by three movements on the X, Y, Z axes. Altitude angle would be depend on the orientation; Maybe a rephrase is needed;
- the usage of position point is not correct! position is enough.
- the experimental part is quite briefly described;
- there are no details related to the mechanical structure and the position error measuring;
- there are no details related to the measurement using the camera;
- the details related to camera calibration and whether this calibration must be performed are not explained;
- there are no details related to the real-time measurement of the terminal effector position;
- the position error and how the measurement was done on the kinematic axes are not described well enough.
Round 2
Reviewer 2 Report
The paper has been improved.